# Gestational Diabetes Melitus and Cord Blood Platelet Function Studied via the PFA-100 System

**DOI:** 10.3390/diagnostics12071645

**Published:** 2022-07-06

**Authors:** Vasiliki Mougiou, Theodora Boutsikou, Rozeta Sokou, Maria Kollia, Serena Valsami, Abraham Pouliakis, Maria Boutsikou, Marianna Politou, Nicoletta Iacovidou, Zoe Iliodromiti

**Affiliations:** 1Neonatal Department, Medical School, National and Kapodistrian University of Athens, 115 28 Athens, Greece; vassouli@hotmail.com (V.M.); theobtsk@gmail.com (T.B.); kolliamaria10@gmail.com (M.K.); boutsikoum@gmail.com (M.B.); niciac58@gmail.com (N.I.); ziliodromiti@yahoo.gr (Z.I.); 2Haematology Laboratory-Blood Bank, Aretaieio Hospital, National and Kapodistrian University of Athens, 115 28 Athens, Greece; serenavalsami@yahoo.com (S.V.); mariannapolitou@gmail.com (M.P.); 32nd Department of Pathology, “Attikon” University Hospital, National and Kapodistrian University of Athens, 124 62 Athens, Greece; apou1967@gmail.com

**Keywords:** neonates, primary hemostasis, gestational diabetes mellitus, PFA-100, platelet function

## Abstract

Neonatal platelet hemostasis, although it has been well described over the recent years, remains elusive in specific patient populations, including neonates from high-risk pregnancies, such as those complicated with gestational diabetes mellitus (GDM). We aimed at evaluating the platelet function of neonates born to mothers with GDM using the platelet function analyzer (PFA-100). Cord blood samples were drawn from each subject and tested with two different agonists to provide two closure time (CT) values (collagen with epinephrine (COL/EPI) and collagen with adenosine diphosphate (COL/ADP)). A total of 84 and 118 neonates formed the GDM and the control group (neonates from uncomplicated pregnancies), respectively. COL/EPI CTs were prolonged in neonates from the GDM group compared to neonates from the control group, while no statistically significant difference of COL/ADP CTs was noted between the two groups, GDM and the control. Higher COL/ADP CTs were demonstrated in neonates born via cesarean section and in neonates with blood group O. A negative correlation between COL/ADP CT and gestational age, white blood cells (WBCs) and von Willebrand factor (VWF) activity was noted in neonates from the GDM group. In conclusion, neonates from the GDM group demonstrate a more hyporesponsive phenotype of their platelets, in comparison to the control neonates.

## 1. Introduction

Gestational diabetes melitus (GDM) refers to the state of hyperglycemia during the second half of pregnancy, with no pre-existing diagnosis of diabetes mellitus, affecting about 14% of pregnancies worldwide [1]. An uncomplicated pregnancy is characterized by insulin resistance, which increases with advancing gestation in order to secure a high glucose supply to the fetus; this may be attributed to hormones and cytokines, including TNFa, IL-6 and adipokines (resistin and leptin) produced by the placenta [2,3]. TNFa is considered as the most important factor that can predict insulin resistance in pregnancy, by impeding insulin actions via impaired intracellular signaling [4]. The adipokines leptin and resistin have also emerged as potential causes of insulin resistance, as they interfere with insulin receptor substrate-1 and lead to its degradation [5]. In GDM, the decreased insulin sensitivity that occurs during pregnancy is exaggerated, due to an alteration of the inflammatory profile; there is excessive production of pro-inflammatory molecules by the placenta, which leads to a hyperglycemic phenotype [6].

GDM-complicated pregnancies are considered high-risk, as they have been linked to adverse outcomes, both maternal and fetal. Maternal complications during pregnancy include spontaneous miscarriage, and pre-eclampsia/pregnancy-induced hypertension; with long-term recurrent GDM in subsequent pregnancies, type 2 diabetes and metabolic syndrome may occur. The offspring of a mother with GDM has immediate and long-term complications, such as macrosomia due to hyperinsulinism, increased risk for trauma at birth, hypoglycemia, hypocalcemia, jaundice, cardiomyopathy, respiratory distress and an increased risk of congenital malformations and type 2 diabetes and metabolic syndrome in the long term. All of these are responsible for the higher mortality and morbidity rates observed in these neonates, which are directly proportional to the maternal glycemic control [7]. High-risk pregnancies, including those complicated with preeclampsia, GDM and autoimmune diseases, have been linked to quantitative changes and in some cases qualitative changes in hemostasis, and especially in platelet function [8]. Hematological indices in these pregnancies have been explored to a substantially higher extent in the mother, leaving several unanswered questions as to how neonatal platelets respond to vascular and metabolic changes [9,10]. Platelet hyper-reactivity was linked to the hyperglycemic and insulin-resistant phenotypes of GDM, as diabetic mothers demonstrate higher MPVs (mean platelet volumes), and thus a more prothrombotic and proinflammatory profile [11,12]. Plateletcrit (a platelet index as a percentage that expresses the volume occupied by platelets in the blood) and inflammatory biomarkers, such as cytokines, were reported to be markedly increased in women with GDM compared with healthy pregnancies [13]. Diabetes mellitus is generally acknowledged as a state of hypercoagulation and platelet hyperaggregation, which is linked to an increased risk of atherosclerosis-related complications [14].

Primary hemostasis in the neonate is a developmental phenomenon, both on a qualitative and on a quantitative level. The fundamental characteristics that have been elucidated in terms of human platelet nature are an age-dependent increase in their number, reaching hemostatic adult values at around full-term gestation, and a hyporeactive phenotype that is counterbalanced by other factors, such as increased hematocrits (Hcts) and von Willebrand factor (VWF) multimers [15,16].

Several functional in vitro tests were used to evaluate the efficiency and examine the parameters of neonatal hemostasis [17]. The platelet function analyzer (PFA-100) is one of them, which is an in-vitro, reproducible system that tests primary hemostasis by quantitatively measuring platelet adhesion, activation, and aggregation. Neonatal closure times (CTs) are universally found to be shorter than those of adults, and this solidifies the model of efficient neonatal hemostasis, with several hematological parameters counterbalancing the otherwise proven intrinsic hypofunctionality of neonatal platelets [18].

In mothers with GDM, a distinct pattern of platelet hyperreactivity is present. Very limited data, however, exists on the effects of the associated glyco-metabolic changes in neonatal platelet function to this day. The present study primarily aims at investigating, via the PFA-100 system, the platelet function of neonates born to mothers with GDM and compares their CTs to that of neonates born from uncomplicated pregnancies. The secondary aims of the study were to obtain correlations of CTs with several perinatal parameters.

## 2. Materials and Methods

### 2.1. Blood Collection and Patients

This is a prospective cohort study of neonates born at Aretaieio Hospital, National and Kapodistrian University of Athens from January 2017 to December 2018. The Hospital Ethics Review Committee approved the study, and the mothers signed an informed consent for participation.

The study population consisted of neonates born to mothers with GDM (GDM group). The offspring of mothers with uncomplicated pregnancies served as the control group. For the GDM group, the inclusion criteria were based on the classification of the American Diabetes Association (2018) that requires testing with an oral glucose tolerance test (OGTT) at 24–28 weeks of gestation [19]. In the cases where OGTT was not performed, abnormal HbA1c values (>6.5%) were taken in consideration. The exclusion criteria included cord blood Hct < 40% or >65%, platelet count < 100 × 10^9^/L, hypothermia (<35 °C), major chromosomal anomaly, family history of bleeding disorder or platelet dysfunction, cord blood pH < 7.25, intrauterine growth restriction (IUGR) and history of pregnancy-induced hypertension.

The demographic data and perinatal parameters of the study population, such as the type of conception, delivery method, gestational age (GA), birth weight, gender, Apgar score, maternal medication during pregnancy (aspirin-ASA, low molecular weight heparins (LMWH), T4 (Thyroxine), peripartum administration of pethidine, epidural/general anesthesia, administration of ampicillin and diabetes medication) and neonatal blood group were recorded. Laboratory data from both the mother (Hct, MPV and platelets) and the neonate (white blood cells, WBCs’ red blood cells, RBCs; neonatal hemoglobin, Hb; Hct; platelets; MPV; VWF activity) were also measured and recorded.

Blood samples were drawn from the umbilical cord vein of the doubly clamped cord, using a 21-gauge needle, and were transferred immediately to plastic tubes containing 3.2% buffered sodium citrate (blood:citrate = 9:1), followed by gentle mixing with the anticoagulant. Once the sample was obtained, Hct and platelet counts were determined and confirmed by the evaluation of peripheral blood smears. Citrated blood was stored at room temperature, and PFA-100 assay was performed within 4 h of sample collection as per the manufacturer’s instructions. All samples were tested with both COL/EPI and COL/ADP cartridges. Each disposable test cartridge has a collagen-coated membrane with an aperture of 147 μm of diameter. The membrane is also coated with either 10 μg epinephrine (COL/EPI cartridge) or 50 μg adenosine diphosphate (COL/ADP cartridge) that serve as platelet agonists. An amount of 0.8 mL of citrate-anticoagulated whole blood was aspirated each time through the aperture under negative pressure, creating high shear stress and simulating real vessel conditions [20]. The incubation of whole blood with either agonist led to platelet activation and the formation of a primary clot. This event marked the final step in the procedure of measuring primary hemostasis and was recorded as CT, the time in seconds needed for the system to form a platelet clot with each agonist [21]. Thus, two CTs were determined for each sample, COL/EPI CT and COL/ADP CT, with the maximum CT measured by PFA-100 being 300 s. If the CTs were >300 s, the results were reported as “non-closure” and all samples containing clots were discarded, thus excluded from the study. An extra aliquot with plasma was preserved at −80 °C for the VWF activity studies.

### 2.2. Statistical Analysis

The statistical analysis was performed by programming in SAS 9.4 for Windows (SAS Institute Inc., Cary, NC, USA) (DiMaggio, 2013; SAS Institute, 2014). In order to evaluate the differences of the measured quantities expressed in a numeric form, we performed the Kruskal–Wallis test, for comparisons or proportions when the quantities were expressed in a qualitative manner (for example normal/abnormal or yes/no values), we applied the chi-square test, and in the case that the expected frequency criteria were not fulfilled, we applied the Fisher exact test. Odds ratios for binary variables were evaluated via the Wald’s *p*-value. Correlations between arithmetic variables were performed with the Spearman correlation coefficient. The statistical significance level was set to 0.05 and all tests were two tailed.

## 3. Results

In total, 202 samples were measured from the neonates, 84 from mothers with diabetes and 118 from the controls. The demographic and laboratory values of the study group and the control population are presented in Table 1. 

Subsequently, the two groups were compared in terms of the various arithmetic and categorical parameters included in the demographic data. There was a lower probability for preterm birth in the control group (20 out of 34 preterm neonates that were included in the study stemmed from the GDM group, which formed a statistically significant difference between the two groups in terms of preterm births, *p* = 0.0253). A higher percentage of diabetic mothers conceived by IVF, compared to the control group (64% of the IVF cases accounted for diabetic mothers, *p* = 0.0433). Both the study group and the control group included neonates of male gender in almost equal numbers. The results of statistical importance were found in the case of Apgar score on the first minute; it was high (>8) for all control neonates, while seven neonates from the GDM group had an Apgar score of <7 on the first minute (8.3% of the group, *p* = 0.0061). The situation was balanced in the Apgar scores of the 5th minute, where no differences were noted, which is the most important parameter predicting the level of difficulty in adaptation in the extrauterine environment. Aspirin administration was higher in the GDM group compared to healthy mothers (*p* = 0.0031). The median neonatal platelet value and median neonatal Hct bared no differences of statistical significance between the two groups

When we compared COL/EPI and COL/ADP CTs between the control and GDM group, COL/EPI CT was found for longer in the GDM group, (median COL/EPI CT was 112.5 s in controls, vs. 129 s in the study group) (*p* = 0.0075). There was no statistically significant difference of COL/ADP CT between the two groups. Those results are depicted in Table 2 and Figure 1.

The correlation of COL/EPI and COL/ADP CT values with various categorical and arithmetic parameters in the GDM group were also performed with a Spearman correlation test, since normality was not ensured.

The impact of delivery mode and neonatal blood group (the only categorical parameters with statistical significance) on CT values are depicted in Table 3. 

COL/ADP CTs were longer in neonates from the GDM group born by CS compared to those born via vaginal delivery (72 s vs. 65 s, *p* = 0.0233). COL/ADP CTs were also longer in neonates from the GDM group with blood group O (73 s vs. 66.5 s for the non-O blood group, *p* = 0.0259). 

The COL/EPI and COL/ADP CT values were also evaluated with reference to maternal medication during pregnancy (aspirin-ASA, LMWH), T4, peripartum administration of pethidine, epidural/general anesthesia, administration of ampicillin), to check if there was a trend for simultaneous increase or decrease. Maternal medication administrated during pregnancy or peripartum did not have any impact on the CT values of neonates from the GDM group. Correlations of COL/EPI and COL/ADP CTs with clinical and other laboratory parameters in the GDM group are depicted in Table 4.

No significant correlations were found between COL/EPI CTs and the recorded neonatal clinical and laboratory parameters of our study, group while negative correlations were found for COL/ADP CTs and gestational age (r = −0.23, *p* = 0.0373). There was a negative correlation of COL/ADP CTs with WBCs (r = −0.34, *p* = 0.0015) and VWF activity (r = −0.36, *p* = 0.0438). Finally, no correlation was found between the PLT counts and either COL/EPI or COL/ADP CTs.

## 4. Discussion

In the present study, a relatively hyporeactive platelet phenotype was demonstrated in neonates born to mothers with GDM, as reflected by the prolonged COL/EPI CTs compared to the control group. 

Hyperinsulinism plays an important role in the pathogenesis of many complications of the neonate of the diabetic mother. Our findings could be partially attributed to insulin’s effect on primary hemostasis. The evidence shows that insulin elicits an anti-thrombotic response via insulin binding on platelet receptors and phosphorylation of insulin receptor substrate 1 (IRS-1). The mechanisms of action comprise lowering agonist-induced platelet aggregation, decreasing calcium mobilization and increasing endothelial nitric oxide (NO) synthesis, since NO is proven to inhibit platelet adhesion and aggregation by increasing cGMP synthesis [22,23].

Another factor that might influence platelet function is the transient hypoglycemia of neonates born to diabetic mothers. More specifically, this pathway has been elucidated in the opposite direction by linking hyperglycemia that is observed during diabetes with platelet aggregation and hyperreactivity. The detected mechanisms for this hyperreactivity are protein kinase C (PKC) activation, as well as oxidative stress elevation and calcium homeostasis alteration. Acute hyperglycemia evokes an aldose reductase-induced polyol accumulation state in the intracellular environment that causes an augmentation in platelet volume and triggers microtubule polymerization, which in turn leads to degranulation and platelet activation [24,25]. Chronic hyperglycemia also induces an increase in calcium influx in platelets and hyperaggregation, which renders the metabolic control issue an important aspect of estimating the prothrombotic risk [26]. It is also known that during the first 72 h of life, almost 50% of all infants of diabetic mothers demonstrate hypocalcemia, due to the abrupt interruption of calcium transfer by the mother and the relatively low levels of parathormone (PTH) and high levels of calcitonin [7,27]. During pregnancy, calcium, as is the case with most nutrients, is diverted from the maternal to the fetal circulation, thus leading to PTH suppression in the fetus, which will take about 72 h to reestablish. This low calcium availability might influence neonatal platelet function, since their activation upon agonist stimulation requires an increase in the intracellular calcium that would not be readily available in the case of offspring of pregnancies complicated with GDM [28].

COL/ADP was slightly prolonged in the GDM group but not in a statistically significant manner. The level of statistical significance appears to depend on the agonist in use, in the sense that platelets of infants of diabetic mothers are more hyporesponsive when exposed in a milieu with epinephrine as an agonist rather than ADP. This could be attributed to external factors that affect agonist availability in each milieu, such as Hct, which is linked to elevated ADP release from the platelets under shear stress conditions. In this case, neonates from the GDM group seem to have higher Hcts than control neonates and this RBC excess affects the otherwise diminished ADP release, as aforementioned, and counterbalances it. Thus, higher Hct may be one factor that compensates for the reduced ADP secretion from neonatal platelets in diabetic pregnancies and bridges the difference between COL/ADP CTs in the two groups [29].

The present study showed that the mode of delivery affected platelet reactivity in neonates of GDM mothers, since COL/ADP CT was longer in CSs compared to vaginal deliveries (*p* = 0.0031). In some cases, higher platelet counts have been found in infants born by vaginal delivery, while others have failed to detect a difference in platelet counts [30,31]. However, regarding platelet activation, the link between mode of delivery and platelet function may lie in the distinct pattern of circulating proinflammatory cytokines that prevail in spontaneous vaginal delivery vs. CS [32]. It has been noted that several proinflammatory cytokines, such as IL-6, IL-1, IL-8 and TNFa, cause platelet hyperactivation and induce thrombopoiesis through direct binding on platelet receptors. The same cytokines that exert a prothrombotic effect are found in higher levels in neonates from vaginal delivery compared to those delivered by CS [33,34].

As far as blood group is concerned, the correlation between blood group O and longer COL/ADP CTs noted in the current study has already been established and has been attributed to the expression of ABO (H) determinants on glycans of human VWF, as well as on platelet surface glycoprotein receptors [21,29]. Individuals with blood group O have 20–30% lower plasma VWF Ag levels than individuals without blood group O [35]. AB antigen expression also plays a role in VWF functional activity, by rendering it more capable to interact with platelets, and group-O VWFs undergo a more rapid proteolysis and cleavage than non-O VWFs. Thus, given the major role of VWF in promoting primary hemostasis by acting as a bridge between platelets and the traumatized endothelium, we may hypothesize that ABO group can influence platelet functionality [36,37].

Our results showed that GA and WBCs measured in cord blood and VWF activity correlated negatively with COL/ADP CTs in the study group. These findings are in line with most previous studies on the effect of perinatal parameters, such as GA, neonatal platelets and VWF activity. The marked hyporeactivity displayed by platelets from preterm neonates in comparison to those from term neonates has been established by several studies and seems to gradually improve over the first 10–14 days of life [38,39]. Platelet adhesion and aggregation have been found to be diminished in preterm neonates, and the negative correlation between CTs and GA has been ascertained by another study only in the case of COL/ADP and not for COL/EPI, which is also the case in our study [40,41]. With severe prematurity, platelet reactivity is affected and neonatal platelets demonstrate a more hyporeactive phenotype, whichis supported by the present study, since most of the preterm neonates showed some of the highest COL/ADP CT values. In our study, the higher percentage of preterm neonates in the GDM group could be a factor that contributes to the CT prolongation noted (neonates from the GDM group accounted for 58.8% of all preterm births, while the controls accounted for 41.2% of them, *p* = 0.0253) This finding was expected, since spontaneous preterm birth is an adverse pregnancy outcome linked with pregnancies complicated with GDM where there is an elevated risk up to 61% for delivery before term especially if macrosomia or fetal distress are present [42,43]. Since pregnant women with GDM tend to deliver before term, if macrosomia or fetal distress is present, these are factors that could lead to higher morbidity [44]. This may explain the higher rate of late preterm infants in our GDM group compared to the control group. 

As for WBCs, the literature has not managed to conclusively link quantitative fluctuations with differences in CTs to date [45]. It is implied that leukocytes may influence platelet function in an indirect manner via prostacyclin synthesis, which inhibits platelet aggregation and would, thus, provoke a prolongation of CTs measured by PFA-100. This is in contradiction to our findings of shorter CTs with higher WBC counts. Evidently, neonatal VWF is a catalytic element of hemostasis, since platelet aggregation and the formation of primary clot depend highly upon its activity. When VWF concentrations are high and ultra-large VWF multimers are present, the intrinsic hypofunctionality of neonatal platelets is counterbalanced and CTs tend to shorten [46,47]. In addition, earlier studies have demonstrated that the most fundamental parameters that can influence PFA-100 CTs are Hct, platelet and WBC count, which all cause a prolongation of COL/EPI and COL/ADP CTs as they tend to decrease [48,49]. In the present study, the lack of correlation between platelet counts and CTs could be attributed to the fact that platelets of our study population fell within normal range.

As far as aspirin administration is concerned, we found that a higher percentage of diabetic mothers received ASA during their pregnancies compared to the control group (*p* = 0.0031). Indeed, several protocols, including ACOG’s guidelines, recommend a low-dose aspirin prophylaxis to limit the risk of preeclampsia in diabetic pregnancies [50]. It is also common knowledge that aspirin inhibits platelet activation and aggregation by blocking COX (cyxlooxygenase), an enzyme responsible for prostaglandin synthesis, thus diminishing PGE2 (prostaglandin E2) and thromboxane 2 (TXA_2_), which are major mediators in the platelet activation- signaling cascade and serve as pro-inflammatory molecules. Studies have revealed that in the context of recent aspirin administration, COL/EPI CT tends to be prolonged, while COL/ADP CT usually is not [51,52]. However, in our study, the higher frequency of aspirin administration in the GDM group did not significantly influence either COL/EPI or COL/ADP CTs as would be expected, and this could be partly attributed to the fact that many of the women in the group had discontinued aspirin uptake more than 7 days before delivery, according to surgical protocols.

This study has several limitations, such as the small size of the population and the fact that they all come from a single center. Our population also included a limited range of premature neonates; hence, to acquire more reliable results, we need to observe a more dispersed sample. Furthermore, for ethical reasons, we did not take venous samples from the neonates; instead, umbilical cord samples were analyzed, which prevented us from corroborating our findings with other neonatal laboratory measurements (glucose, calcium, hematological variables). Moreover, given the fact that our data referred to cord blood samples, the results of the study should be interpreted with caution, especially in the case of neonates that have other risk factors for bleeding events, such as prematurity, where the prolonged CTs could be considered as a marker of platelet hypofunctionality and lead to interventions such as transfusion-guided therapy. Another limitation was the lack of samples for measuring VWF activity deriving from the control group, which was due to a lack in reagents. Finally, we lacked samples that could be used in the direction of inflammatory markers’ measurement, which would help to corroborate our findings with the already established proinflammatory profile of GDM pregnancies.

## 5. Conclusions

In conclusion, our data suggest that platelets from neonates born to mothers with GDM tend to be hyporesponsive compared to those born from mothers with uncomplicated pregnancies, as CTs are overall higher in the study group compared to the control group. However, this discrepancy does not seem to induce an increased risk for bleeding, since both COL/EPI and COL/ADP CTs fall within the normal range for the neonatal population, as recently published by our research group, where a large homogeneous population of healthy term and preterm neonates was evaluated with PFA-100 and the range for COL/EPI and COL/ADP CTs were found to be 101–178 s and 64–86 s, respectively [18]. We also found that conditions such as CS, several medications, as well as neonatal hematological parameters, especially platelets, GA, WBCs and VWF activity, can influence CTs, by causing prolongation and hyporesponsiveness in the case of CS and lower GAs, WBCs and VWF activity.

PFA-100 is a system that serves to predict hematological disorders and closely monitor neonates that are at high risk for bleeding complications and require no more than a small cord-blood sample, which is usually readily available. Thus, with regard to measuring CTs via PFA-100 in neonates from high-risk pregnancies (such as GDM, preeclampsia, IUGR), any prolongation may warrant the need for closer observation to prevent or manage appropriately any hemorrhagic events that may lead to the emergence of neurologic or cardio-respiratory sequelae that could compromise the prognosis or even increase mortality.

## Figures and Tables

**Figure 1 diagnostics-12-01645-f001:**
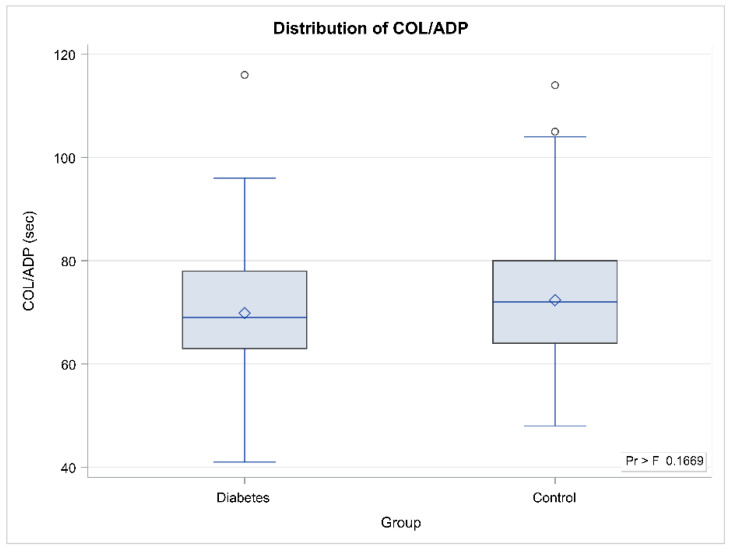
Box and whisker plots for COL/EPI CT (above) and COL/ADP CT (below) between the study group (*n* = 84) and the control group (*n* = 118). Whisker limits show minimum and maximum values (excluding outliers that appear as circles), 25th and 75th percentile are the box lower and upper parts, while median and mean values appear as the horizontal line inside the box and the diamond symbol, respectively.

**Table 1 diagnostics-12-01645-t001:** Demographic and laboratory data of neonates born to mothers with GDM and control group.

Type of Data	Variable		Control Group *n* = 118	GDM Group *n* = 84	*p* Value
**Gestational data**	Preterm		14 (11.86%)	20 (23.8%)	0.025
	IVF		6 (5.08%)	11 (13.09%)	0.043
	CS		78 (66.1%)	62 (44.29%)	0.242
	Forced delivery		12 (10.16%)	2 (2.38%)	0.737
	Pethidine		5 (4.23%)	4 (4.76%)	0.830
	Anesthesia (epidural)		89 (75.42%)	64 (76.19%)	0.900
	Anesthesia (general)		1 (0.85%)	4 (4.76%)	0.078
	Ampicillin peripartum		12 (10.17%)	5 (5.95%)	0.322
**Neonatal characteristics**	Gender	Male	55 (46.61%)	48 (57.14%)	0.140
		Female	63 (53.39%)	36 (42.86%)	
	Apgar 1′ < 5		0 (0%)	1 (1.19%)	0.006
	Apgar 1′ 5–7		0 (0%)	6 (7.14%)	
	Apgar 1′ 8–10		118 (100%)	77 (91.67%)	
	Apgar 5′ 5–7		0 (0%)	1 (1.19%)	
	Apgar 5′ 8–10		118 (100%)	83 (98.8%)	0.235
	Gestational age (weeks)		39.07 (38.28–39.84)	39 (37.14–39.7)	0.101
	Birth weight (g)		3305 (3080–3560)	3190 (2690–3560)	0.080
	Birth weight percentile		50 (35–69)	40.5 (19–73.5)	0.044
	Birth temperature		36.3 (36.1–36.6)	36.2 (36–36.5)	0.341
	Neonatal blood group O		46 (38.98%)	30 (35.71%)	0.683
**Maternal medication**	Aspirin < 7 days before delivery		2 (1.69%)	11 (13.1%)	0.003
	Aspirin > 7 days before delivery		8 (6.78%)	8 (9.52%)	
	LMWH		10 (8.47%)	8 (9.52%)	0.796
	Diabetes treatment: diet		0 (0%)	56 (66.67%)	0.0000
	Diabetes treatment: insulin		0 (0%)	23 (27.38%)	
	Diabetes treatment: pills		0 (0%)	2 (2.38%)	
	Diabetes treatment: none		0 (0%)	3 (3.57%)	
**Neonatal hematologic parameters**	Neonatal WBCs (/μL)			12350 (10,200–15,200)	
	Neonatal Hct (%)		45 (42.2–47.7)	46.55 (43–49.85)	0.065
	Neonatal PLTs (×10^9^/L)		251 (207.5–296)	260 (215.5–303)	0.502
	Neonatal MPV (fl)		9.8 (8.4–10.5)	9.5 (8.25–10.25)	0.331
	Neonatal VWF activity			119.05 (97.4–137.7)	
**Maternal hematologic parameters**	Maternal Hct (%)		36 (34–38)	37 (34–39)	0.199
	Maternal PLTs (×10^9^/L)		199 (173–234)	207 (180–260.5)	0.150
	Maternal MPV (fl)		10.9 (10.4–11.8)	11.3 (10.7–12)	0.092

Abbreviations: CS: cesarean section, IVF: in vitro fertilization, LMWH: low molecular weight heparin, Hct: hematocrit, WBC: white blood cells, MPV: mean platelet volume, VWF: von Willebrand factor, PLT platelets. Footnotes: arithmetic data are presented as median values (25th–75th percentile); categorical characteristics are presented in the form of frequencies and the relevant percentages (number of occurrences/either group, control or GDM).

**Table 2 diagnostics-12-01645-t002:** Comparison of COL/EPI and COL/ADP CTs between neonates born to mothers with GDM and control group.

Variable	Levels	*n*	COL/EPI (Median, IQR)	*p*-Value	COL/ADP (Median, IQR)	*p*-Value
**Group**	**Diabetes**	84	129 (100.5–164)	0.0075	69 (63–78)	0.168
	**Control**	118	112.5 (93–145)		72 (64–80)	

Footnotes: showing the group of comparison along with the COL/EPI and COL/ADP median levels, followed by the q1–q3 ranges (25th–75th centile) and the *p*-values.

**Table 3 diagnostics-12-01645-t003:** The comparison of CT values in the GDM group, with respect to delivery mode and neonatal blood group.

Variable	Levels	*n*	COL/EPI CT (Median, IQR)	*p*-Value	COL/ADP CT (Median, IQR)	*p*-Value
**Delivery mode**	**Cesarean Section**	62	129 (100–161)	0.680	72 (64.5–78.5)	0.023
	**Vaginal delivery**	22	132 (101–192)		65 (60–69)	
**Neonatal blood group**	**O**	30	137.5 (103–193)	0.3801	73 (66–79)	0.026
	**Non-O**	53	127 (99–155)		66.5 (60.5–74)	

Footnotes: showing the group of comparison along with the COL/EPI and COL/ADP median levels, followed by the q1–q3 ranges (25th–75th centile) and the *p*-values.

**Table 4 diagnostics-12-01645-t004:** Correlation of COL/EPI and COL/ADP CTs with clinical and other laboratory (arithmetic) parameters of neonates from GDM mothers.

Variable	COL/EPI CT	COL/ADP CT
**Gestational age**	0.09575	−0.230
	0.3862	0.037
**Neonatal WBCs**	0.09523	−0.345
	0.3889	0.002
**Neonatal VWF activity**	−0.26577	−0.364
	0.1415	0.044
**PLTs**	−0.02279	−0.180
	0.837	0.105

Abbreviations: WBCs: white blood cells, VWF: von Willebrand factor, PLT: platelets. Footnotes: data are presented as correlation coefficient indices (rho), and respective *p* values. The nonparametric Spearman’s correlation test was used for the statistical evaluation.

## Data Availability

The data presented in this study are available on request from the corresponding author.

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
