# Peer review of "Gestational Diabetes Melitus and Cord Blood Platelet Function Studied via the PFA-100 System"

_diagnostics, 2022, doi:10.3390/diagnostics12071645_

Round 1

Reviewer 1 Report

The authors presented a relatively simple but useful study to demonstrate platelet function of neonates born to mothers with gestational diabetes melitus compared to control study group along with studying correlations of closure time values with several perinatal parameters. I would like to suggest a few things in order for the data presented and discussed to be clearer.

Line 59-62: Since the authors mention that hematological indices several high-risk pregnancies including GDM has been studied substantially it would be good to include a couple of reference here for anyone that wants to review this information.

Line 134-136: This sentence does not add to the actual materials or methods.

Table 1:

- In the section for “Neonatal Characteristics”, “Apgar 8-10” should probably be “Apgar 5’ 8-10”

Line 161-179: Are the authors using this paragraph to highlight data from Table 1? If so, a suggestion to the authors would be to use the information that is provided in Table 1. In most instances, it is difficult to match up with the information provided in the paragraph and Table 1.  Another suggestion would be to provide the information in the order that it appears in the table to make it easier for the readers to follow.

For example:

·      Line 161-162: If the study group contained both male and female neonate then what is the reason for showing only “Gender-Male” in the section for “Neonatal Characteristics” in Table 1?

·      Line 167: “The median neonatal platelet value was 256000 and median neonatal Hct was 45.9%.”..for control or GDM group? Also, where are these values on the Table 1?

·      Line 174-176: The sentence “There was a lower probability for preterm birth in the control group” makes sense looking at data in Table 1. However, the additional information provided following the sentence in parenthesis is confusing (not sure where these numbers are coming from?)

Line 178-179: “A higher percentage of diabetic mothers conceived by IVF, compared to the control group”…this is also indicated in Table 1. However, this data was not discussed at all in the “Discussion”. If the authors could explain if IVF has any impact on GDM mothers and their neonates of GDM mothers and their platelet level and function, also if there is any published work in this matter to discuss this then it would be good. Since this is one of the areas where any significant difference is seen in this data the reader would be eager to know more about it.

Line 184: Make the “f” capitalized in “figure 1” so it is consistent with the rest.

Line 200: Make the “t” capitalized in “table 3” so it is consistent with the rest.

Line 206: Make the “t” capitalized in “table 4” so it is consistent with the rest.

Line 236: close the parenthesis after p=0.0438.

Line 233-239: Does this not affect the GDM maternal platelet? If the mothers have hyperinsulinism then should their platelet be hyporesponsive too? Did the authors come across any data to study that touches on this topic? It may be useful include those references in this discussion.

Line 274-275: The citations need to be edited to [26,27].

Line 291: The citations need to be edited to [30.31].

Line 293-295: This statement can benefit from some references.

Line 324: Consider editing “hasn’t” to has not.

Line 332-335: Although common knowledge for some, this statement can benefit from some references.

Line 353: Since in this paper, von Willebrand Factor was abbreviated as VWF, the authors may want to edit the “vWF” to VWF.

Line 376-377: There was a link for supplemental figure/table but could not access it. Also did not see the mention of the supplemental material in the article.

Regarding references…References are commonly seen before the period (.) in sentences. In this article, it is seen mostly after the period. Is this okay with the editorial board? If not, may need to edited.

Author Response

Reviewer 1 comments:

The authors presented a relatively simple but useful study to demonstrate platelet function of neonates born to mothers with gestational diabetes melitus compared to control study group along with studying correlations of closure time values with several perinatal parameters. I would like to suggest a few things in order for the data presented and discussed to be clearer.

 We thank you for your input in our manuscript.

Line 59-62: Since the authors mention that hematological indices several high-risk pregnancies including GDM has been studied substantially it would be good to include a couple of reference here for anyone that wants to review this information.

References added to elucidate this point:

Janes, S.L. and A.H. Goodall, Flow cytometric detection of circulating activated platelets and platelet hyper-responsiveness in pre-eclampsia and pregnancy. Clin Sci (Lond), 1994. 86(6): p. 731-9.

Gioia, S., et al., Gestational diabetes: is it linked to platelets hyperactivity? Platelets, 2009. 20(2): p. 140-1.

Line 134-136: This sentence does not add to the actual materials or methods.

This sentence has been removed from materials and methods.

Table 1:

- In the section for “Neonatal Characteristics”, “Apgar 8-10” should probably be “Apgar 5’ 8-10”

The above segment has been corrected.

Line 161-179: Are the authors using this paragraph to highlight data from Table 1? If so, a suggestion to the authors would be to use the information that is provided in Table 1. In most instances, it is difficult to match up with the information provided in the paragraph and Table 1.  Another suggestion would be to provide the information in the order that it appears in the table to make it easier for the readers to follow.

For example:

  • Line 161-162: If the study group contained both male and female neonate then what is the reason for showing only “Gender-Male” in the section for “Neonatal Characteristics” in Table 1?
  • Line 167: “The median neonatal platelet value was 256000 and median neonatal Hct was 45.9%.”..for control or GDM group? Also, where are these values on the Table 1?
  • Line 174-176: The sentence “There was a lower probability for preterm birth in the control group” makes sense looking at data in Table 1. However, the additional information provided following the sentence in parenthesis is confusing (not sure where these numbers are coming from?)

Table 1 has been modified so as to be more clear with respect to “gender”, the order of appearance of the information has been modified according to the order of appearance in the table and the additional information about preterm births has been changed.

Line 178-179: “A higher percentage of diabetic mothers conceived by IVF, compared to the control group”…this is also indicated in Table 1. However, this data was not discussed at all in the “Discussion”. If the authors could explain if IVF has any impact on GDM mothers and their neonates of GDM mothers and their platelet level and function, also if there is any published work in this matter to discuss this then it would be good. Since this is one of the areas where any significant difference is seen in this data the reader would be eager to know more about it.

To our knowledge no studies have been carried out on the effect of IVF on GDM mothers or platelet function/level, except for the knowledge that GDM is a more frequent complication of IVF pregnancies compared to unassisted. We thank you the reviewer for your observation, however given the limited data regarding the impact of IVF on the hemostatic profile of either mothers or their neonates and the small sample size of IVF pregnancies in our study population, we did not proceed to further evaluation of this finding. Additionally the impact of IVF on the primary hemostasis of neonates is part of our running project.

Line 184: Make the “f” capitalized in “figure 1” so it is consistent with the rest.

The correction has been made.

Line 200: Make the “t” capitalized in “table 3” so it is consistent with the rest.

The correction has been made.

Line 206: Make the “t” capitalized in “table 4” so it is consistent with the rest.

The correction has been made.

Line 236: close the parenthesis after p=0.0438.

The correction has been made.

Line 233-239: Does this not affect the GDM maternal platelet? If the mothers have hyperinsulinism then should their platelet be hyporesponsive too? Did the authors come across any data to study that touches on this topic? It may be useful include those references in this discussion.

Mothers with GDM do not present with hyperinsulinism as the major pathogenetic mechanism of GDM is insulin resistance. Thus insulin’s antithrombotic effect is hindered in GDM mothers’ platelets.

Line 274-275: The citations need to be edited to [26,27].

The correction has been made.

Line 291: The citations need to be edited to [30.31].

The correction has been made.

Line 293-295: This statement can benefit from some references.

 Reference has been added: O'Donnell, J., & Laffan, M. A. (2001). The relationship between ABO histo-blood group, factor VIII and von Willebrand factor. Transfusion medicine (Oxford, England)11(4), 343–351.

Line 324: Consider editing “hasn’t” to has not.

The correction has been made.

Line 332-335: Although common knowledge for some, this statement can benefit from some references.Reference added: Katz, J. A., Moake, J. L., Mcpherson, P. D., Weinstein, M. J., Moise, K. J., Carpenter, R. J., & Sala, D. J. (1989). Relationship Between Human Development and Disappearance of Unusually Large von Willebrand Factor Multimers From Plasma (Vol. 73, Issue 7). www.bloodjournal.org

Line 353: Since in this paper, von Willebrand Factor was abbreviated as VWF, the authors may want to edit the “vWF” to VWF.

The correction has been made.

Line 376-377: There was a link for supplemental figure/table but could not access it. Also did not see the mention of the supplemental material in the article.

 The link for supplemental material has been omitted.

Regarding references…References are commonly seen before the period (.) in sentences. In this article, it is seen mostly after the period. Is this okay with the editorial board? If not, may need to edited.

The correction has been made, references have been placed before the period.

Reviewer 2 Report

Reviewer comments:

Comments to the Author

This manuscript aimed to evaluate the platelet function of neonates born to mothers with gestational diabetes mellitus (GDM) using Platelet Function Analyzer (PFA-100). Authors reported that COL/EPI CTs were prolonged in neonates from the GDM group compared to neonates from the control group. Neonates from the GDM group demonstrate a more hyporesponsive phenotype of their platelets, in comparison to control neonates.

The aim of the manuscript is very interesting, but the authors are needed to improvise this manuscript with substantial experiments to confirm their findings on Comparison of COL/EPI and COL/ADP CTs. Most part of the manuscript is well written, and discussion of results were postulate according to the evidence provided. Authors should try to incorporate some inflammatory markers in the samples tested.

Major criticisms

• Authors need to compare other inflammatory markers in the samples collected from the patients to support their finding on comparison of COL/EPI and COL/ADP CTs.

• Please include the information in the figure legend on the number of samples analyzed for Figure 1.

• Please undergo a thorough check of the manuscript for typographical and grammatical errors.

Author Response

Reviewer 2 comments:

This manuscript aimed to evaluate the platelet function of neonates born to mothers with gestational diabetes mellitus (GDM) using Platelet Function Analyzer (PFA-100). Authors reported that COL/EPI CTs were prolonged in neonates from the GDM group compared to neonates from the control group. Neonates from the GDM group demonstrate a more hyporesponsive phenotype of their platelets, in comparison to control neonates.

The aim of the manuscript is very interesting, but the authors are needed to improvise this manuscript with substantial experiments to confirm their findings on Comparison of COL/EPI and COL/ADP CTs. Most part of the manuscript is well written, and discussion of results were postulate according to the evidence provided. Authors should try to incorporate some inflammatory markers in the samples tested.

We thank you the reviewer for your valuable input on our manuscript.

Major criticisms

  • Authors need to compare other inflammatory markers in the samples collected from the patients to support their finding on comparison of COL/EPI and COL/ADP CTs.

Unfortunately, no samples were kept to be analyzed for other inflammatory markers (CRP, IL-1/6) and corroborate our findings with a higher pro-inflammatory profile of GDM pregnancy. Hence this has now been mentioned as a limitation of the present study.

  • Please include the information in the figure legend on the number of samples analyzed for Figure 1.

The correction has been made.

  • Please undergo a thorough check of the manuscript for typographical and grammatical errors.

The manuscript has been double checked by a native speaker of the English language and modifications have been made.

Round 2

Reviewer 2 Report

The answers from the authors showed that there were limitation in the study designing and planning to confirm their findings, which is very essential to conclude the results. 

Authors need to find alternatives to plan few experiments or include some samples/cohort to do these experiments.

Author Response

“The answers from the authors showed that there were limitation in the study designing and planning to confirm their findings, which is very essential to conclude the results. 

Authors need to find alternatives to plan few experiments or include some samples/cohort to do these experiments.”

We thank the reviewer for the useful and very relevant comments on our study.

The relationship between Gestational Diabetes Mellitus (GDM) (and most types of diabetes) and an inflammatory profile that is higher than pregnancy’s baseline has already been established by several studies. [1] The link between low-grade inflammation and a prothrombotic state in GDM has also been investigated via Mean Platelet Volume (MPV), that serves as a marker for a biologically more active phenotype of platelets and has been associated to higher levels of insulin resistance during pregnancy, thus predicting or even aiding to the follow-up of GDM.[2] Inflammation could lead to an MPV elevation since increased cytokine production (like IL-6) has been linked to platelet activation and additionally polymorphonuclear leukocytes have been found to release PAF (platelet activating factor) [3, 4]

Platelet hyperreactivity in GDM is deemed to be a multifactorial phenomenon, with several factors other thaninflammation contributing to the activation of primary hemostasis in a diabetic milieu. Hence, GDM causes damage to the vascular endothelium not only through the effects of inflammatory factors but also through oxidative stress and high glucose concentrations/ advanced glycation end products (AGEs).[5] Also, insulin seems to play an important role in primary hemostasis given the fact that insulin receptors are present on platelets and insulin resistance hinders intracellular signal transduction and thus blunts the antithrombotic effect of insulin on platelets. [6-8] Calcium homeostasis is another field of interest that connects diabetes with platelet hyper-functionality. [9]

All in all, it was beyond the scope of this study to investigate the specific link between inflammation and hemostasis in GDM. So based on the data of the aforementioned studies that have already been carried out and reported the impact of diabetes mellitus and GDM on primary hemostasis we conducted our study with an aim of investigating, via the PFA-100 system, the platelet function of neonates born to mothers with GDM and compare their CTs to those of neonates born from uncomplicated pregnancies. Our data suggests that there is an impact of GDM on neonatal platelet function and although we find the reviewer’s suggestion to associate haemostatic disorders with increased inflammation a very interesting idea, it could not be incorporated in this study which is part of one of the author’s thesis. However, this provides the ground for further and more thorough investigations and serves as a trigger to create new pathways for further research on the possible pathophysiological mechanisms of this connection and constitute our future study project.

References:

  1. Pantham, P., I.L. Aye, and T.L. Powell, Inflammation in maternal obesity and gestational diabetes mellitus. Placenta, 2015. 36(7): p. 709-15.
  2. Zhou, Z., et al., Mean Platelet Volume and Gestational Diabetes Mellitus: A Systematic Review and Meta-Analysis. J Diabetes Res, 2018. 2018: p. 1985026.
  3. Burstein, S.A., et al., Cytokine-induced alteration of platelet and hemostatic function. Stem Cells, 1996. 14 Suppl 1: p. 154-62.
  4. Sisson, J.H., et al., Production of platelet-activating factor by stimulated human polymorphonuclear leukocytes. Correlation of synthesis with release, functional events, and leukotriene B4 metabolism. J Immunol, 1987. 138(11): p. 3918-26.
  5. He, Y. and N. Wu, Research Progress on Gestational Diabetes Mellitus and Endothelial Dysfunction Markers. Diabetes Metab Syndr Obes, 2021. 14: p. 983-990.
  6. Hers, I., Insulin-like growth factor-1 potentiates platelet activation via the IRS/PI3Kalpha pathway. Blood, 2007. 110(13): p. 4243-52.
  7. Kaur, R., M. Kaur, and J. Singh, Endothelial dysfunction and platelet hyperactivity in type 2 diabetes mellitus: molecular insights and therapeutic strategies. Cardiovasc Diabetol, 2018. 17(1): p. 121.
  8. Strauss, T., et al., Impaired Platelet Function in Neonates Born to Mothers with Diabetes or Hypertension During Pregnancy. Klinische Pädiatrie, 2010. 222: p. 154-7.
  9. Ferreira, I.A., et al., IRS-1 mediates inhibition of Ca2+ mobilization by insulin via the inhibitory G-protein Gi. J Biol Chem, 2004. 279(5): p. 3254-64.

Round 3

Reviewer 2 Report

Authors response is acceptable and the manuscript can be accepted.